# A Light-Weight Autoregressive CNN-Based Frame Level Transducer Decoder for End-to-End ASR

Hyeon-Kyu Noh and Hong-June Park *

Department of Electronic and Electrical Engineering, Pohang University of Science and Technology (POSTECH), Pohang 37673, Republic of Korea; nohhg92@postech.ac.kr
* Correspondence: hjpark@postech.ac.kr

**Abstract:** A convolutional neural network (CNN) transducer decoder was proposed to reduce the decoding time of an end-to-end automatic speech recognition (ASR) system while maintaining accuracy. The CNN of 177 k parameters and a kernel size of 6 generates the probabilities of the current token at the token level, at the token transition of the output token sequence. Two probabilities of the current token, one from the encoder and the other from the CNN are added to the frame level to reduce the decoding step to the number of input frames. An encoder composed of an 18-layer conformer was combined with the proposed decoder for training with the Librispeech data set. The forward-backward algorithm was used for training. The space and re-appearance tokens are added to the 300-word piece tokens to represent the token string. A space token appears at a frame between two words. A comparison with the autoregressive decoders such as transformer and RNN-T decoders demonstrates that this work provides comparable WERs with much less decoding time. A comparison with non-autoregressive decoders such as CTC indicates that this work enhanced WERs.

**Keywords:** speech recognition; autoregressive speech recognition; end-to-end; CNN; transducer decoder

## 1. Introduction

Currently, end-to-end automatic speech recognition(ASR) methods are widely used to extract text from the time-domain speech input [1–3], because the entire end-to-end ASR block can be trained by using a deep learning network. The end-to-end ASR block consists of an encoder and a decoder. For example, the encoder converts the mel-frequency spectrum coefficient (MFSC) speech input ([$4 * T_{enc}$, 83]) to an intermediate output ([$T_{enc}$, 302]); $4 * T_{enc}$ is the number of frames of the input speech sentence. Each frame has an overlapped window of 25 ms long and proceeds in 10 ms steps. The encoder output ([$T_{enc}$, 302]) represents the probabilities of 302 tokens in 40 ms steps; the 302 token includes 300 word pieces and two special tokens. The decoder converts the encoder output to a text ([$U$]) at the token level; $U$ is the number of word pieces corresponding to the encoder input.

The encoder block of the end-to-end ASR can be implemented with various deep learning networks, such as convolutional neural network (CNN) [4–6], long short term memory (LSTM) [7,8] or transformer [9–11].

The decoder block can be implemented either in autoregressive or non-autoregressive methods. The autoregressive decoder accepts the previous tokens along with the encoder output as the decoder input. The non-autoregressive decoder accepts only the encoder output as the decoder input.

The connectionist temporal classification(CTC) [12] is one of the representative non-autoregressive decoders; it generates a loss function to train the end-to-end ASR which generates a text at the token level from the speech input in the frame level. The frame level refers to the time domain, that is, variables in the frame level can change values in unit time intervals, in this work, in 40 ms intervals. The token level refers to the text domain, that is, variables in the token level can change values at the token transition. The CTC

decoder does not include any trainable neural network blocks. The CTC performs two functions. One is the conversion of the encoder output ($[T_{enc}, 301]$) to the text ($[U]$). The CTC selects a max-valued index out of 301 tokens per frame to generate an intermediate token sequence ($[T_{enc}]$) and eliminates the blank tokens and the consecutively duplicate tokens from the intermediate token sequence to generate the $[U]$ text. The blank token represents a 'non-token' frame or a 'token transition' frame. The other is to generate a loss function to train the encoder block by accumulating all the matching probabilities between the intermediate token sequence ($[T_{enc}]$) and the ground truth text ($[U]$) along both the forward(previous) and the backward(future) directions. Compared to the autoregressive decoders, the CTC-based decoders [13–17] have merits of much less computation time with less accuracy, because there are no trainable parameters and the parallel computation is enabled by no feedback operation.

The representative autoregressive decoders for end-to-end ASR are the transformer decoder [9] and the RNN transducer(RNN-T) decoder [18]. The RNN-T decoder is similar to the CTC decoder except that it adds the processed previous token sequence ($[U, 300]$) and the encoder output ($[T_{enc}, 300]$) to generate the decoder input ($[U + T_{enc}, 300]$). An LSTM block is included in the RNN-T decoder to generate the processed token sequence from the previous token sequence ($[U]$).

The transformer decoder includes a transformer block to convert the encoder output ($[T_{enc}, 256]$) to the probabilities of the output token sequence ($[U, 302]$); this enables the use of a simpler loss function of cross entropy. However, the transformer decoder suffers from long computation time because it cannot employ parallel computation due to the feedback operation and the relatively large number of trainable parameters. Besides, the accuracy is degraded in the transformer decoder without using the additional beam search. (Table 1) compares the characteristics of the above-mentioned decoders used for end-to-end ASR.

**Table 1.** Comparison of decoders for end-to-end ASR.

|  | Autoregressive | Decoding Level [Length] | Special Tokens | Training Algorithm |
|---|---|---|---|---|
| Transformer [9] | O | token level [$U$] | \<sos\>, \<eos\> | Cross-Entropy |
| CTC [12] | X | frame level [$T_{enc}$] | blank | Forward/backward |
| RNN-T [18] | O | Token + Frame [$U + Tenc$] | blank | Forward/backward |
| **This work** | **O** | **frame level [$T_{enc}$]** | **space, re-appearance** | **Forward/backward** |

The transformer decoder generates the token probabilities at the token level from the encoder output and the previous token sequence by using the transformer blocks. Since the number ($[U]$) of these token probabilities is the same as the number ($[U]$) of the ground truth tokens, the transformer decoder can calculate the loss by using the cross entropy without any special tokens such as blank tokens. Only two special tokens(\<sos\>, \<eos\>) are employed in the transformer decoder to align the attention blocks at the start and end points of an input utterance.

The CTC decoder generates the token probabilities at the frame level which are the encoder output. Since the number ($[T_{enc}]$) of these token probabilities is not the same as the number ($[U]$) of the ground truth token probabilities, the CTC decoder cannot use the cross entropy but uses the forward/backward algorithm for the loss computation. The forward/backward algorithm adds "blank" tokens as token separators and at non-token frames for training with the mismatch between $T_{enc}$ and $U$.

The RNN-T decoder works similarly to the CTC-decoder except that it generates the token probabilities at the frame + token level from the encoder output and the previous token sequence. Thus, the number of these token probabilities is $T_{enc} + U$.

This work generates the token probabilities at the frame level to reduce the decoding time. Since the number ($[T_{enc}]$) of these token probabilities is not the same as the number

($[U]$) of the ground truth token probabilities, the forward/backward algorithm is used in this work, as in the CTC and RNN-T decoders. The word separator token ("space token") is used in this work instead of the blank token to avoid the deletion of short-duration tokens in the CTC decoders. Since the blank token is inserted at every token boundary and occupies a frame time in the CTC decoder, a short-duration token that lasts only one or two frames is deleted at the CTC decoder output. Because the space token of this work occurs less frequently than the blank token, the short-duration token is less probable to be deleted in this work. The blank token occupies around 70 percent of frames in the trained frame level token sequence of the CTC decoder while the space token occupies around 50 percent of frames in the trained frame level token sequence of this work. The re-appearance token is introduced in this work to replace the role of the blank token which keeps the consecutive-same token sequence.

In this work, an autoregressive decoder without using the beam search is proposed for end-to-end ASR; the autoregressive architecture enhances accuracy and the non-use of the beam search reduces the computation time. To further reduce the computation time, two schemes are proposed in this work. One is to add the token probabilities at the frame level ($[T_{enc}]$) as mentioned before. The other is the use of a small-size CNN in the decoder feedback loop. Compared to the RNN-T and the transformer decoders, this work achieved comparable accuracy with half and 1/30 decoding times, respectively.

Section 2 explains the proposed CNN-based frame level autoregressive decoder for end-to-end ASR. Section 3 presents the experimental results. Section 4 discusses this work.

## 2. Models

The end-to-end ASR of this work consists of an encoder and the proposed decoder. The encoder accepts the MFSC input of an utterance ($[4 * T_{enc}, 83]$) in 10 ms steps and generates the token probabilities of the utterance ($[T_{enc}, 302]$) at each frame. Each frame proceeds in 40 ms steps. The encoder input includes 80 MFSC and 3 pitch information [19] (pitch frequency and its first and second time derivatives). The 302 tokens of the encoder output include 300 word pieces [20,21] and two special tokens (space token, re-appearance token); the space token ("/") separates words and the re-appearance token ("~") represents the same token is repeated at the token level [22]. For example, "$/ffeeee \sim\sim\sim bblelele/$" at the frame level is translated into "feeble" at the token level.: "$le$" is a word piece in this example.

Table 2 tabulates the number of frames where the blank and the space token occur at the token probabilities of the trained CTC decoder, and this works for the Librispeech dataset, including the train-clean-100 and dev-clean. The blank token dominates the frame space by occupying around 70% of the trained CTC decoder, while the space token occupies less frame space (about 50%) in the trained model of this work.

**Table 2.** The number of blank tokens in CTC and space token in this work at the frame level for LibriSpeech dataset.

|  | Train-Clean-100 (%) | Test (%) |
|---|---|---|
| CTC (blank) | 5,986,615 (66.7) | 1,283,175 (68.1) |
| This work (space) | 4,119,724 (45.9) | 888,293 (47.1) |

While the blank token supports as many consecutively same tokens as possible in the CTC decoder, the re-appearance token of this work works correctly for only up to two consecutively same tokens and misses the third consecutively same tokens. In the ground-truth token level LibriSpeech dataset used in this work, only 18 utterances have three consecutive same tokens out of the total 7,117,087 tokens (Table 3), and no utterances have four consecutively same tokens.

**Table 3.** The number of consecutive same token sequences at the token level of the LibriSpeech dataset (the total number of tokens = 7,117,087).

| | Train-Clean-100 | dev-Clean | dev-Other | Test-Clean | Test-Other | Sum (%) |
|---|---|---|---|---|---|---|
| 2 consecutive same tokens | 72,555 | 1410 | 1219 | 1311 | 1104 | 77,599 (1.1%) |
| 3 consecutive same tokens | 16 | 0 | 0 | 1 | 1 | 18 (0.0003%) |

The encoder shown in Figure 1, is made up of a sub-sampling CNN, an 18-layer conformer [23], and a linear layer. This encoder is used with the RNN-T decoder, the transformer decoder and the CTC decoder as well as this work, for comparison.

The decoder shown in Figure 2 accepts the token probabilities at the frame level (encoder output) ($[T_{enc}, 302]$) as input and generates the token sequence ($[U]$) as output during the inference operation. Because the autoregressive models are more accurate than the non-autoregressive models such as CTC, the autoregressive model is adopted in this work. The autoregressive decoder of this work adds two token probabilities to generate the 302 added probabilities at each frame. One of the two token probabilities is the frame-level token probability generated from the MFSC speech input by the encoder, and the other is the token-level token probability generated from the previous token sequence. Then, the decoder selects the token with the largest added probability as the current frame at each frame and appends the current token to the output token sequence at the token transition, that is, when the current token is different from the token at the preceding frame. The decoder repeats the above operations sequentially in frame units and hence the output token sequence ($[u]$) grows as the frame proceeds.

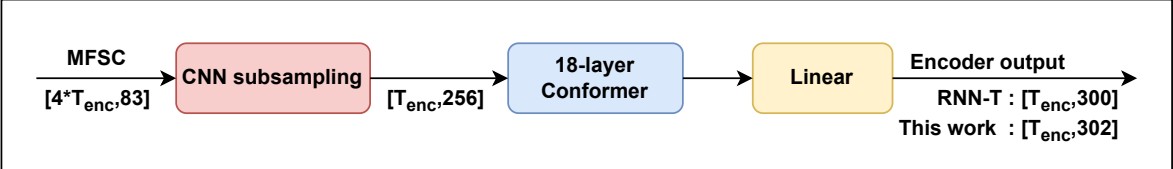

**Figure 1.** Encoder used in this work.

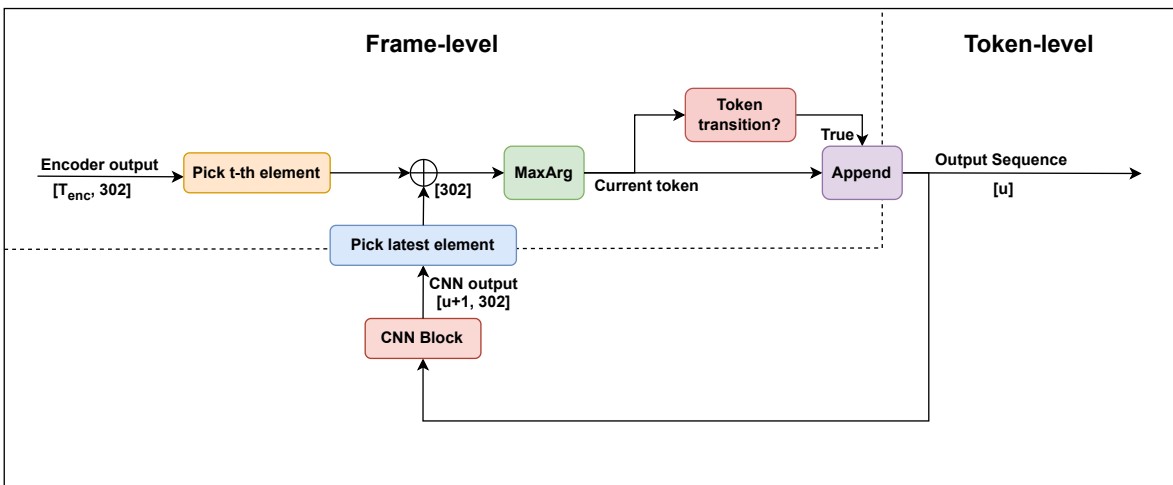

**Figure 2.** Proposed frame level autoregressive decoder (inference).

### 2.1. Inference

The CNN block shown in Figure 3 accepts the previous output token sequence as input and generates the token probabilities as output at the token level. Since the CNN block observes only the previous output token sequence, it works as a causal system. Because

the CNN block has a 1D CNN layer with a kernel size of 6, it generates the current token probabilities based on the six previous output tokens. Since the average number of word pieces of a word is 2.2 (Librispeech [24] test other, 300 word pieces), six tokens correspond to 2.73 words. This is equivalent to a 3.73 g language model, which is close to the 4-gram language model that has been reported to optimal size [25].

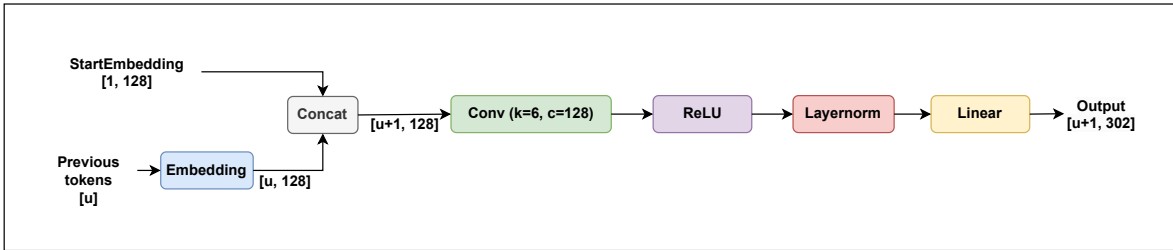

**Figure 3.** The CNN block of the proposed decoder.

The RNN-T decoder yields the best accuracy among the decoders without using the beam search. The RNN-T decoder is different from this work in two aspects. One is that the RNN-T decoder adds the token probabilities in the frame + token level ($[T_{enc} + L]$) while this work adds in the frame level ($[T_{enc}]$) (Figure 4) This increases the decoding steps from $T_{enc}$ to $T_{enc} + L$ in the RNN-T decoder and hence increases the decoding time. While this work accepts the added token probabilities once at each frame ($[T_{enc}]$), to select the current token with the largest added probability, the RNN-T decoder accepts the added token probabilities whenever either the frame-level token probability($[T_{enc}]$, encoder output) or the token-level token probability ($[U]$) changes.

The other is that the RNN-T decoder uses an LSTM layer while a CNN layer is used in this work. Because the number of trainable parameters is around 1M in the RNN-T decoder and 177 k in this work and the decoding steps are larger in the RNN-T decoder, the decoding time of the RNN-T decoder is around 1.7 times of the RNN-T decoder.

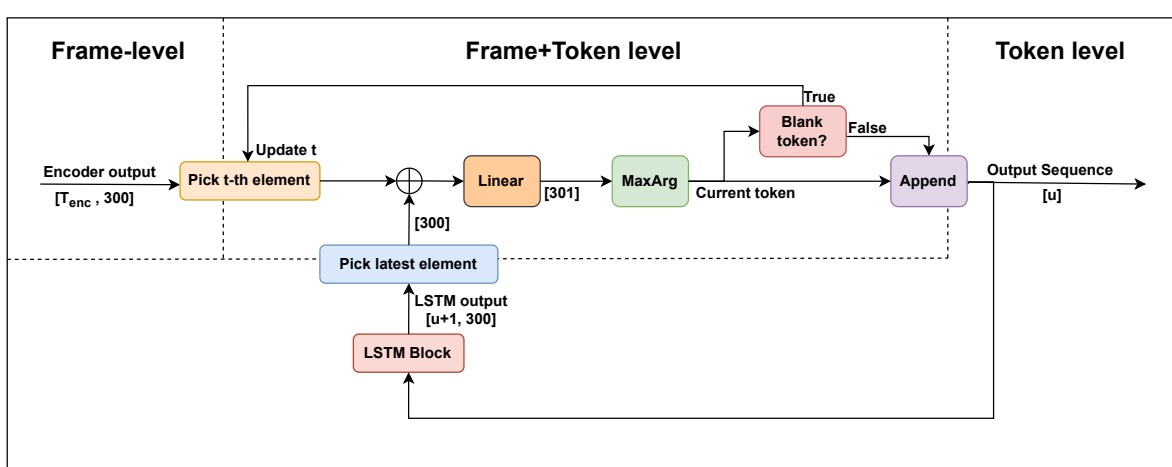

**Figure 4.** Proposed frame-level autoregressive decoder (inference).

*2.2. Training*

The encoder and the CNN block of the proposed decoder are trained using the model shown in Figure 5, which is similar to the training model of the RNN-T decoder [18]. The loss function is minimized to match the input sequence in MFSC to the ground truth token sequence y [$U$]:

$$Loss = \log \sum_{u=1}^{U} \alpha(t, u)\beta(t, u) \tag{1}$$

The loss function is determined by the sum of the products of $\alpha(t, u)$ and $\beta(t, u)$ for all $u$ from 1 to $U$, where the product of $\alpha(t, u)$ and $\beta(t, u)$ represents the probability that the token at the frame $t$ is $y(u)$. $y$ is the ground truth token sequence and $U$ is the number of tokens in $y$. $\alpha(t, u)$ is called the forward variable and $\beta(t, u)$ is called the backward variable [12]:

$$\alpha(t, u) = \sum_{\pi \in B^{-1}(y), \pi_t = y_u} \prod_{t'=1}^{t'=t} p_{t', u'}(\pi_{t'}) \tag{2}$$

$$\beta(t, u) = \sum_{\pi \in B^{-1}(y), \pi_t = y_u} \prod_{t'=t+1}^{t'=T_{enc}} p_{t', u'}(\pi_{t'}) \tag{3}$$

$B^{-1}(y)$ is the set of frame-level token sequences that generate the token-level ground truth $y$ by deleting repeating tokens. $p_{t,u}(k)$ in (2) and (3) is determined by the sum of the frame level probability $x_t(k)$ from the encoder output and the token level probability $z_u(k)$ from the CNN output, for a token $k$:

$$p_{t,u}(k) = softmax(x_t(k) + y_u(k)) \tag{4}$$

The forward variable $\alpha(t, u)$ can be calculated recursively from the start $(t = 1, u = 1)$:

$$\alpha(t, u) = \alpha(t-1, u-1)p_{t,u-1}(y_u) + \alpha(t-1, u)p_{t,u}(y_u) \tag{5}$$

Similarly, the backward variable $\beta(t, u)$ can be calculated recursively from the end $(t = T_{enc}, u = U)$:

$$\beta(t, u) = \beta(t+1, u+1)p_{t+1,u}(y_{u+1}) + \beta(t+1, u)p_{t+1,u}(y_u) \tag{6}$$

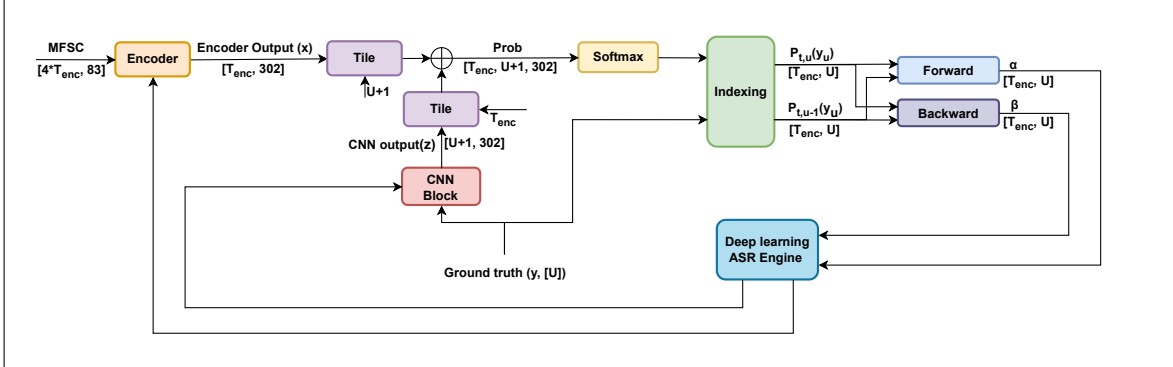

**Figure 5.** The proposed frame level autoregressive decoder (training).

The recursive equations of this work (5), (6) are similar to those of CTC [12] because both decoders work at the frame level, except that the CTC decoder includes a "Blank" token while this work does not. The forward path computation of $\alpha(t, u)$ was presented graphically in Figure 6 for this work, CTC and RNN-T decoders. Both forward and backward path computation equations of $\alpha(t, u)$ and $\beta(t, u)$ are compared in Table 4 for the preceding three decoders.

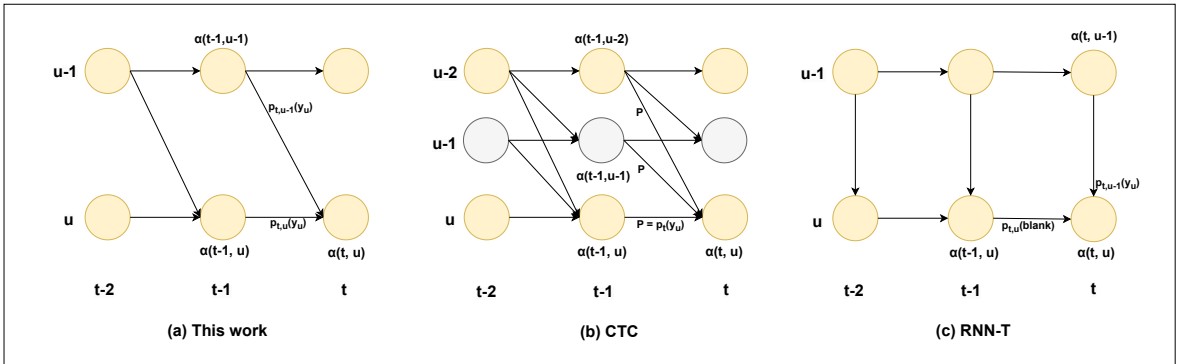

**Figure 6.** Forward path computation of $\alpha(t, u)$ of decoders for end-to-end ASR.

**Table 4.** Forward/Backward algorithm of decoders for end-to-end ASRs.

| Decoder Type | | Forward Variable $\alpha(t, u)$ | Backward Variable $\alpha(t, u)$ |
|---|---|---|---|
| CTC | wordpiece (even $u$) | $(\alpha(t-1, u-2) + \alpha(t-1, u-1) + \alpha(t-1, u)) * p_t(y_u)$ | $(\beta(t-1, u-2) + \beta(t-1, u-1) + \beta(t-1, u)) * p_t(y_u)$ |
| | blank (odd $u$) | $(\alpha(t-1, u-1) + \alpha(t-1, u)) * p_t(y_u)$ | $(\beta(t-1, u+1) + \beta(t-1, u)) * p_t(y_u)$ |
| RNN-T | | $\alpha(t-1, u) * p_{t-1,u}(blank) + \alpha(t, u-1) * p_{t-1,u}(y_u)$ | $\beta(t+1, u) * p_{t,u}(blank) + \beta(t, u+1) * p_{t,u}(y_u)$ |
| This work | | $\alpha(t-1, u-1) * p_{t+1,u-1}(y_u) + \alpha(t-1, u) * p_{t,u}(y_u)$ | $\beta(t+1, u+1) * p_{t+1,u}(y_{u+1}) + \beta(t+1, u) * p_{t,u}(y_u)$ |

The partial derivatives of Loss w.r.t. $x_t$ (encoder output) and $z_u$ (CNN output) are used to compute the gradients for backpropagation:

$$\frac{\partial Loss}{\partial x_t} = -\frac{1}{\exp(-Loss)} \{ \sum_u \frac{\partial \alpha(t, u) \beta(t, u)}{\partial p_{t,u}(y_u)} \frac{\partial p_{t,u}(y_u)}{\partial x_t} \} \tag{7}$$

$$\frac{\partial Loss}{\partial z_u} = -\frac{1}{\exp(-Loss)} \{ \sum_t \frac{\partial \alpha(t, u) \beta(t, u)}{\partial p_{t,u}(y_u)} \frac{\partial p_{t,u}(y_u)}{\partial z_u} + \sum_t \frac{\partial \alpha(t, u+1) \beta(t, u+1)}{\partial p_{t,u}(y_{u+1})} \frac{\partial p_{t,u}(y_{u+1})}{\partial z_u} \} \tag{8}$$

### 3. Experiments and Results

The CNN block of the proposed decoder was trained along with the encoder by using the 100 h Librispeech [24] clean dataset (train-clean-100). To avoid overfitting, data augmentation techniques such as speed perturbation [26] and SpecAugment [27] were used for the training. For the speed perturbation, training was performed three times for each dataset, one at $0.9\times$ , another at $1.0\times$ and the other at $1.1\times$ speed. For the SpecAugment, two frequency masks and two time masks were applied to every input utterance along with the time warping. Each frequency mask ranges a consecutive range of up to 30 frequency bins. Each time mask ranges from consecutive time frames up to 40 frames. The time warping shifts the time frame up to five frames toward left and right. The 300 word pieces generated from SentencePiece [20] and two special tokens were used in this work to represent the token sequence. The special tokens include the space token and the re-appearance token. The space tokens are used for no-token frames or to separate words in this work, while the blank tokens of the CTC decoder are used to separate tokens or for no-token periods. The re-appearance token is used for the frames where the same token is repeated in the token sequence. The Adam optimizer [23] was used for the training

with the hyperparameters of $\beta_1 = 0.9, \beta_2 = 0.98, \epsilon = 10^{-9}$. Also, the Noam learning rate scheduling [9] was used with the 25k warmup steps and the learning rate factor of 5.0. The model was trained up to 120 epochs, and the trainable parameters of the last 10 epochs were averaged to obtain the inference model. The kernel size is 15 for the CNN blocks of the 18-layer conformer comprising the encoder block shown in Figure 1.

To assess the performance of this work, the WERs and the numbers of trainable parameters of this work were compared in Table 5 with the non-autoregressive decoders (CTC [12], self-conditioned CTC [17], intermediate CTC [16]) and the autoregressive decoders (transformer [9], RNN-T [18]); the same encoder with 30M trainable parameters was used in this comparison. This work gives better WERs compared to the non-autoregressive decoders and comparable WERs to the autoregressive decoders while the number of trainable parameters is much smaller than those of the autoregressive decoders (177 k in this work versus 10 M in the transformer and 1 M in RNN-T). To train the RNN-T decoder, the dropout rate was 0.3 for the encoder and 0.2 for the decoder. To train the 6-layer transformer decoder, the dropout rate was reduced to 0.1 to alleviate the influence of the dropout on the attention operation, and the CTC was included in the loss function to enhance accuracy [28,29]. In this work, the dropout rate was 0.1 for the encoder and 0.3 for the decoder. The training codes were based on the ESPNet [30].

**Table 5.** Comparison of WER (%) and number of trainable parameters.

| Model | | Params (M) | Word Error Rate (%) | |
|---|---|---|---|---|
| | | | Test-Clean | Test-Other |
| Auto-regressive | **This work** | **0.18** | **6.8** | **18.4** |
| | Transformer [9] | 9.6 | 7.2 | 18.0 |
| | RNN-T [18] | 1.0 | 6.6 | 18.3 |
| Non-Autoregressive | CTC [12] | 0 | 7.7 | 20.7 |
| | Self-conditioned CTC [17,21] | 0 | 6.9 | 19.7 |
| | Intermediate CTC [16,21] | 0 | 7.1 | 20.2 |

Also, In Table 6 the decoding times are compared along with the encoding times during the inference operation for the Librispeech test-other dataset (2939 sentences, around 5.1 h long); the AMD EPYC 7402P 24-Core Processor CPU was used for this inference operation. The proposed decoder works 30.9 and 1.65 times faster than the transformer and the RNN-T decoders, respectively, while the CTC decoder is 13.5 times faster than this work. In this work, the encoder spends 7.6 times longer time than the decoder.

However, this work takes longer training time than other decoders due to the less optimized gradient computation step. The gradient computation equations (Equations (7) and (8)) of this work have the computational complexity of $O(T_{enc} * U)$ while those of CTC, RNN-T and transformer decoders have $O(T_{enc})$, $O(T_{enc} + U)$ and $O(U)$, respectively. It is expected that this long training time may be reduced by sharing intermediate computation results in future works.

**Table 6.** Comparison of decoding times in seconds during inference using LibriSpeech test other data set and training times in hours using Librispeech train-clean-100.

| Model | Inference Time (s) | | | Training Time (h) |
|---|---|---|---|---|
| | Encoding Time | Decoding Time | Relative Decoding Time | |
| **This work** | **368** | **48.6** | **1.0** | **75.9** |
| Transformer [9] | 350 | 1500 | 30.9 | 41.7 |
| RNN-T [18] | 365 | 80.1 | 1.65 | 38.1 |
| CTC [12] | 371 | 3.63 | 0.074 | 24.6 |

The kernel size of the CNN block of the decoder was set to 6 in this work because the best WER was observed empirically at the kernel size 6 in the range from 4 to 24 for four datasets (Librispeech dev-clean, dev-other, test-clean, test-other) as shown in Figure 7; the number of the layers and the channels of the CNN block was set to 1 and 128, respectively. The kernel size 6 corresponds to 2.73 words on average; this corresponds to the 3.73-gram language model, which is close to the known-to-be-optimum 4-gram language model [25].

The optimum number of the layers and the channels of the CNN block was found empirically to be 1 and 128, respectively, with the kernel size set to 6 (Table 7). With the increase in the number of trainable parameters of the CNN block, the WER degrades due to the overfitting of the potentially wrong previous token sequence with the beam search not used in this work.

**Table 7.** Comparison of WER with different layers and channels of CNN in this work.

| CNN Spec | | | # Trainable Params | WER (%) | | | |
|---|---|---|---|---|---|---|---|
| Kernel | Layers | Channels | | Dev-Clean | Dev-Other | Test-Clean | Test-Other |
| 6 | 1 | 96 | 115 k | 6.6 | 18.4 | 7.0 | 18.7 |
| **6** | **1** | **128** | **177 k** | **6.4** | **18.3** | **6.8** | **18.4** |
| 6 | 1 | 192 | 340 k | 6.4 | 18.3 | 6.9 | 18.8 |
| 6 | 2 | 128 | 276 k | 6.5 | 18.6 | 6.9 | 18.8 |

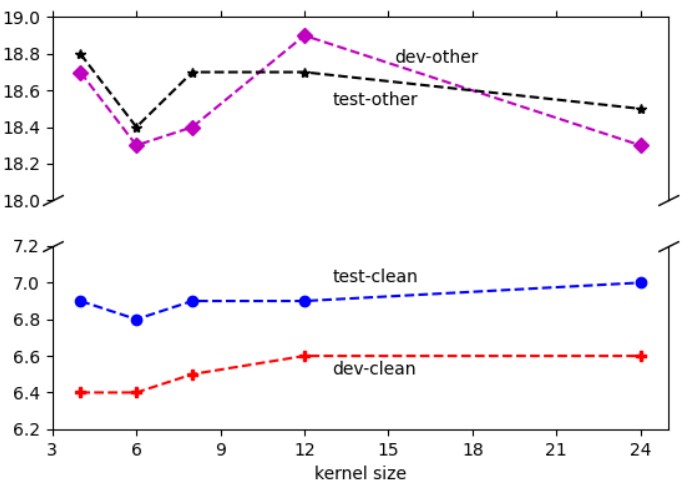

**Figure 7.** WERs of this work versus the kernel size of CNN ( red +: dev clean, blue O: test clean, purple ◇: dev other, black *: test other).

The RNN-T decoder provides the best WER among the decoders for the end-to-end ASRs without using the beam search. There are two differences between this work and the RNN-T decoder. One is the decoding step is $[T_{enc}]$ in this work and $[T_{enc} + U]$ in the RNN-T decoder. The other is that a CNN block with 177 k trainable parameters is used in this work, while an LSTM block with 910 k parameters and a linear block with 90 k parameters is used in the RNN-T decoder. To assess the contribution of the two differences to the decoding time, the number of trainable parameters, and WER, the LSTM block of the RNN-T decoder was replaced by the CNN block of this work. Table 8 shows the comparison of the three models and presents that the change from the LSTM to CNN ("RNN-T(LSTM → CNN)" in Table 8) significantly reduces the number of trainable parameters, from 1 M to 270 k with the comparable WERs and the decoding time. The additional change of the decoding steps from $[T_{enc} + U]$ to $[T_{enc}]$ reduces the decoding time by 1.65 times with the comparable WERs.

**Table 8.** Comparison of this work with RNN-T decoder.

| Model | Relative Decoding Time | Params (M) | Word Error Rate (%) | |
|---|---|---|---|---|
| | | | Test-Clean | Test-Other |
| Original RNN-T (LSTM) | 1.65 | 1.0 | 6.6 | 18.3 |
| RNN-T (LSTM → CNN) | 1.83 | 0.27 | 6.8 | 18.3 |
| This work (CNN) | 1.0 | 0.18 | 6.8 | 18.4 |

Table 9 shows that applying the beam search to the decoder output usually enhances WER with a large increase in the decoding time. However, the beam search could not enhance WERs significantly in this work; it is conjectured that this may be due to the small-size CNN of the decoder in this work, while large-size decoders could help improve WERs in other decoders(transformer, RNN-T).

**Table 9.** WERs and decoding times with and without beam search, beam size = 10.

| | Word Error Rate (%) | | | | | |
|---|---|---|---|---|---|---|
| | Beam Size = 1 | | | Beam Size = 10 | | |
| | Relative Time | Test-Clean | Test-Other | Relative Time | Test-Clean | Test-Other |
| **This work** | **1.0** | **6.8** | **18.4** | **24.14** | **6.8** | **18.3** |
| RNN-T | 1.65 | 6.6 | 18.3 | 30.36 | 6.3 | 17.9 |
| Transformer | 30.9 | 7.2 | 18.0 | 347.77 | 6.7 | 17.8 |

## 4. Discussion

A frame-level autoregressive decoder is proposed to reduce the decoding time for the inference operation of end-to-end ASRs while maintaining good WERs. This was achieved by adopting the autoregressive architecture and reducing the decoding steps and the number of trainable parameters of the decoder. To reduce the decoding steps, the decoder is operated at the frame level, that is, the decoder generates the probabilities of the token set once in each frame, at every 40 ms in this work. To reduce the number of trainable parameters, only a CNN block of 177 k parameters was employed in the proposed decoder of this work, while the transformer decoder has a 6-layer transformer with 9.6 M trainable parameters and the RNN-T decoder has an LSTM block (910 k parameters) and a linear block (90 k parameters). Two special tokens (space, re-appearance) are used along with 300-word piece tokens to represent the token string. The proposed frame-level autoregressive decoder was trained along with an encoder with 18 conformers (30 M parameters), by using the 100 h Librispeech dataset (train-clean). A comparison of the proposed decoder with the state-of-the-art autoregressive decoders combined with the same encoder used in this work demonstrates that the proposed decoder provides comparable WERs to other decoders with much less decoding time (1.7 and 30.9 times less than the RNN-T and the transformer decoders, respectively). A comparison with non-autoregressive decoders presents that the proposed decoder gives better WERs than other decoders with a longer decoding time.

**Author Contributions:** Conceptualization, H.-K.N. and H.-J.P.; software, H.-K.N.; project administration, H.-K.N. and H.-J.P. worked together during the whole editorial process of the manuscript. All authors were involved in the preparation of this manuscript. All authors have read and agreed to the published version of the manuscript.

**Funding:** This work was supported by the National Research Foundation of Korea (NRF) grant funded by the Korea government (MSIT) (No. 2022R1A2C2003451).

**Institutional Review Board Statement:** Not applicable.

**Informed Consent Statement:** Not applicable.

**Data Availability Statement:** The raw data supporting the conclusions of this article will be made available by the authors on request.

**Conflicts of Interest:** The authors declare no conflicts of interest.

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
