# Peer review of "A Light-Weight Autoregressive CNN-Based Frame Level Transducer Decoder for End-to-End ASR"

_applsci, doi:10.3390/app14031300_

Round 1

Reviewer 1 Report

Comments and Suggestions for Authors

This paper present an autorregresive method to reduce decoding time in an Automatic Speech Recognition (ASR) system, while maintaining the accuracy, measured in terms of WER. This is an improvement in performance, but not new in terms of novelty or better recognition rate, which is also about 10% more accurate than non-autorregresive methods, which are faster methods. However, the methods are well presented and explained in the paper.

I would like to suggest some improvements:

- Proposed method uses data augmentation techniques to avoid overfitting, details of the parameters used for data augmentation seems to be missing in the paper.

- Figure 7 shows the error metric versus the kernel size, a critical parameter. It's not clear to me the explanation of obtained results. A value of 6 in kernel size seems to be better, but why WER is worse when kernel size increase with the clean dataset, while it seems to decrease with the other dataset. An interpretation of those results, in the opinion of authors, would be interesting.

Reviewer 2 Report

Comments and Suggestions for Authors

This paper presents A light-weight autoregressive convolutional neural network (CNN) based frame-level transducer decoder for end-to-end Automatic Speech Recognition. The authors used a convolutional neural network transducer decoder to reduce the decoding time of end-to-end automatic speech recognition. The authors used a CNN of 177k parameters with a kernel size of 6 for the generation of probabilities of the current token at a sequence. The authors added the probabilities of the current token in the frame-level to reduce the decoding step to number of input frames. The comparison is made with the autoregressive decoders such as transformer and RNN-T decoders and a comparison is made with the non-autoregressive decoders such as CTC.

Reviewer 3 Report

Comments and Suggestions for Authors

You can better support existing knowledge in the introduction and specialization in explaining the functionality and the model, which later is an important part of the discussion and comparison with the proposed model.

The article, in my opinion, is well-written and methodical, and the division into Auto-regressive and Non-Autoregressive helps the reader a lot.

You must like to mention the possible clinical applications of the model.

Reviewer 4 Report

Comments and Suggestions for Authors

This manuscript proposed a light-weight autoregressive CNN-based frame-level transducer decoder for end-to-end ASR. Comparison with the autoregressive decoders such as transformer and RNN-T decoders demonstrates that this work provides comparable WERs with much less decoding time. Comparison with the non-autoregressive decoders such as CTC indicates that this work enhanced WERs. Overall, this manuscript is well written. However, there is room for improvement in terms of innovation and the persuasiveness of the conclusions. 

1. The methods used in the paper are all existing methods, and there is only optimization in the specific CNN modeling process, so the innovation is not clear enough.

2. The manuscript does not provide detailed explanations on how the parameters were determined during the training process of the CNN model.

3. In Section 3, the introduction of the specific data processing methods and analysis process is not detailed enough.

Round 2

Reviewer 4 Report

Comments and Suggestions for Authors

This manuscript can be accepted now.